# GABAergic inhibition in human hMT+ predicts visuo-spatial intelligence mediated through the frontal cortex

Yuan Gao[1†], Yong-Chun Cai[2†], Dong-Yu Liu[1,3], Juan Yu[1,3], Jue Wang[1], Ming Li[4], Bin Xu[1], Tengfei Wang[2], Gang Chen[5], Georg Northoff[6], Ruiliang Bai[7*], Xue Mei Song[1,3*]

[1]Department of Neurosurgery of the Second Affiliated Hospital, Interdisciplinary Institute of Neuroscience and Technology, Zhejiang University School of Medicine, Hangzhou, China; [2]Department of Psychology and Behavioral Sciences, Zhejiang University, Hangzhou, China; [3]Key Laboratory of Biomedical Engineering of Ministry of Education, Qiushi Academy for Advanced Studies, College of Biomedical Engineering and Instrument Science, Zhejiang University, Hangzhou, China; [4]College of Intelligence Science and Technology, National University of Defense Technology, Changsha, China; [5]University of Ottawa Institute of Mental Health Research, University of Ottawa, Ottawa, Canada; [6]Oujiang Laboratory (Zhejiang Lab for Regenerative Medicine, Vision and Brain Health), Hangzhou, China; [7]MOE Frontier Science Center for Brain Science & Brain-Machine Integration, Zhejiang University, Hangzhou, China

*For correspondence:
ruiliangbai@zju.edu.cn (RB);
songxuemei@zju.edu.cn (XMS)

†These authors contributed equally to this work

**Abstract** The prevailing opinion emphasizes fronto-parietal network (FPN) is key in mediating general fluid intelligence (gF). Meanwhile, recent studies show that human MT complex (hMT+), located at the occipito-temporal border and involved in 3D perception processing, also plays a key role in gF. However, the underlying mechanism is not clear, yet. To investigate this issue, our study targets visuo-spatial intelligence, which is considered to have high loading on gF. We use ultra-high field magnetic resonance spectroscopy (MRS) to measure GABA/Glu concentrations in hMT+ combining resting-state fMRI functional connectivity (FC), behavioral examinations including hMT+ perception suppression test and gF subtest in visuo-spatial component. Our findings show that both GABA in hMT+ and frontal-hMT+ functional connectivity significantly correlate with the performance of visuo-spatial intelligence. Further, serial mediation model demonstrates that the effect of hMT+ GABA on visuo-spatial gF is fully mediated by the hMT+ frontal FC. Together our findings highlight the importance in integrating sensory and frontal cortices in mediating the visuo-spatial component of general fluid intelligence.

## eLife assessment

This **important** study adopts a comprehensive approach: functional connectivity, biochemistry, and psychophysics to reveal a holistic understanding of the relationship between GABA-ergic inhibition in the human MT+ region and visuo-spatial intelligence. The evidence supporting the conclusion is **convincing**. The result advances our understanding of how the human MT+ is assemble into complex cognition as an intellectual hub, and will be of interest to researchers in psychology, cognitive science, and neuroscience.

**Figure 1.** Hypothesis and experimental design. (**a**) Schematic of hypothesis. The inhibition mechanism centered on MT+ GABA, including the molecular level: the GABAergic inhibition in MT+ (cyan circle), brain connectivity level: hMT+-frontal functional connectivity (blue circle), and behavior level: hMT+ specific surround suppression of visual motion (red circle), contributes to the visuo-spatial component of general fluid intelligence (gF) (3D domain, yellow circle). (**b**) Schematic of experimental design. Session 1 (rectangle box of short line) was the functional MRI and MRS scanning at resting state. Session 2 (rectangle box of solid line) was another region of MRS acquisition. In the two sessions, the order of MRS scanning regions (hMT+ and early visual cortex [EVC] [primarily in V1]) was counterbalanced across participants. There was a structural MRI scanning before each MRS data acquisition. The interval between the two sessions was used for behavioral measurement (rectangle box of dotted line): block design task (BDT) and psychophysical task-motion discrimination. Sold lines indicate the experiment sequence.

## Introduction

General fluid intelligence (gF) is a current problem-solving ability, which shows high inter-individual differences in humans (*Cattell, 1963*). At the beginning of the last century, *Spearman, 1904*, proposed that some general or g factor contributes to our gF. One key component of gF is visuo-spatial intelligence, usually tested by visual materials, shows high g-loading (*Colom et al., 2006*; *Deary et al., 2010*; *Jung and Haier, 2007*). The exact neural mechanisms of the interplay of visuo-spatial intelligence with gF remain yet unclear, though.

The 'neuro-efficiency' hypothesis is one explanation for individual differences in gF (*Haier et al., 1988*). This hypothesis puts forward that the human brain's ability to suppress irrelevant information leads to more efficient cognitive processing. Correspondingly, using a well-known visual motion paradigm (center-surround antagonism; *Liu et al., 2016*; *Tadin et al., 2003*), Melnick et al. found a strong link between suppression index (SI) of motion perception and the scores of the block design test (BDT, a subtest of the Wechsler Adult Intelligence Scale [WAIS]), which measures the visuo-spatial component (3D domain) of gF (*Melnick et al., 2013*). Motion surround suppression (SI), a specific function of human extrastriate cortical region, middle temporal complex (hMT+), aligns closely with this region's activities (*Gautama and Van Hulle, 2001*). Furthermore, hMT+ is a sensory cortex involved in visual perception processing (3D domain; *Cumming and DeAngelis, 2001*). These findings suggest that hMT+ potentially plays a significant role in 3D visuo-spatial gF by facilitating the efficient processing of 3D visual information and suppressing irrelevant information. However, more evidence is needed to uncover how the hMT+ functions as a core region for 3D visuo-spatial intelligence.

Frontal cortex is usually recognized as the cognitive core region (*Duncan et al., 2000*; *Gray et al., 2003*). Strong connectivity between the cognitive regions suggests a mechanism for large-scale information exchange and integration in the brain (*Barbey, 2018*; *Cole et al., 2012*). Therefore, the potential conjunctive coding may overlap with the inhibition and/or excitation mechanism of hMT+. Taken together, we hypothesized that 3D visuo-spatial intelligence (as measured by BDT) might be predicted by the inhibitory and/or excitation mechanisms in hMT+ and the integrative functions connecting hMT+ with frontal cortex (*Figure 1a*).

To investigate our hypothesis, this work conducted multi-level examination including biochemical (glutamatergic-GABAergic in hMT+), regional-systemic (brain connectivity with hMT+-based), and

behavioral (visual motion function in hMT+) levels to reveal if hMT+ contributes to the 3D visuo-spatial component of gF. We employ ultra-high field (7T) magnetic resonance spectroscopy (MRS) technology to reliably resolve GABA and Glu concentrations (*Ende, 2015*; *Liu et al., 2022*; *Song et al., 2021*). To verify the specificity of hMT+, we used early visual cortex (EVC, primarily in V1)-based GABA/Glu as control as it mediates the 2D rather than 3D visual domain (*Cumming and DeAngelis, 2001*).

Our findings first demonstrate that GABAergic inhibition mechanisms (but not excitatory Glu) in hMT+ region relate to 3D visuo-spatial ability. Further, analysis of functional brain connectivity at rest reveals that the network (between MT+ and frontal cortex) relating to MT+ GABA and perceptual suppression contribute to the visuo-spatial intelligence. Our results provide direct evidence that inhibitory mechanisms centered on GABA levels in MT+ region (a sensory cortex) mediate multi-level visuo-spatial component (3D domain) of gF thus drawing a direct connection of biochemistry, brain connectivity, and behavior.

## Results

To determine whether the function of hMT+ cortex contributes to visuo-spatial component (3D domain) of gF, we adopted the experimental design depicted in *Figure 1b*. Thirty-six healthy subjects participated in this study. Participants underwent two MRI sessions: the first encompassing resting-state fMRI and MRS, and the second solely involving MRS. A 30 min interval separated these sessions, during which participants performed motion discrimination tasks (using center-surround antagonism stimuli; *Tadin et al., 2003*) and the BDT, which assesses the visuo-spatial ability (3D domain) of gF

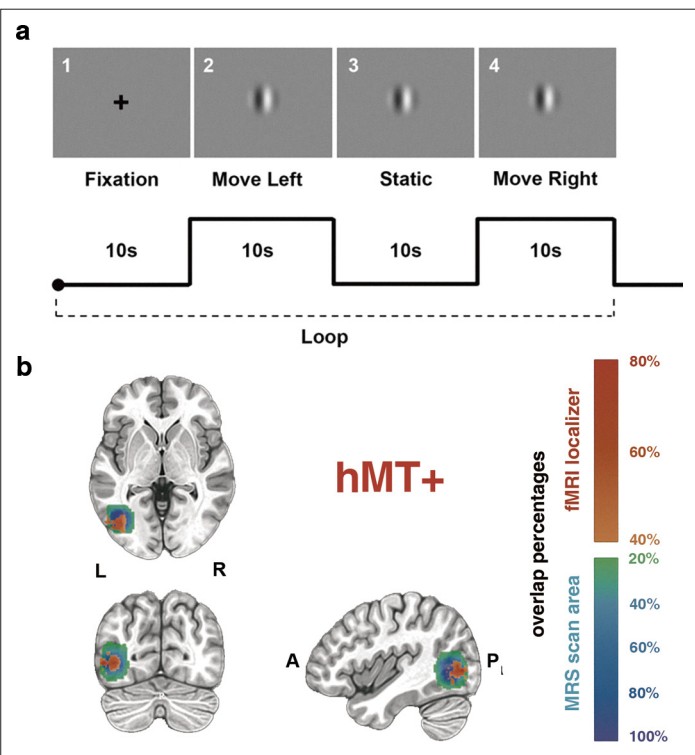

**Figure 2.** hMT+ localizer scans and hMT+ magnetic resonance spectroscopy (MRS) volume-of-interest (VOI) placement. (**a**) Single task block designs. First: a cross fixation on the center of the screen (10 s). Second: a moving grating (2°) toward left last 10 s. Third: the grating keeps static for 10 s. Fourth: the grating moves toward right last 10 s. The localizer scans consist of 8 blocks. (**b**) hMT+ location and MRS VOI placement. The upper template is the horizontal view. The lower templates from left to right are coronal and sagittal views. The warm color indicates the overlap of fMRI activation of hMT+ across 14 subjects, the cold color bar indicates the overlap of MRS VOIs across all subjects.

The online version of this article includes the following figure supplement(s) for figure 2:

**Figure supplement 1.** The left hemisphere early visual cortex (EVC) (primarily in V1) magnetic resonance spectroscopy (MRS) scanning ROI.

(*Fangmeier et al., 2006*). In the motion discrimination tasks, a grating of either large or small size was randomly presented at the center of the screen. The grating drifted either leftward or rightward, and participants were asked to judge the perceived moving direction. While in the BDT, participants were asked to rebuild the figural pattern within a specified time limit using a set of red and white blocks. Both the volume-of-interests (VOIs) of MRS scanning in the left hMT+ (targeted brain area) and the left EVC (primarily in V1, control brain area) had dimensions of $2\times2\times2$ cm$^3$, and the MRS scanning sequences were randomized across the two sessions. The hMT+ MRS VOIs were demarcated using an anatomical landmark (*Dumoulin et al., 2000*). For 14 subjects, we also utilized fMRI to functionally pinpoint the hMT+ to validate the placement of the VOI (*Figure 2a and b*). The EVC (primarily in V1) MRS VOIs (*Figure 2—figure supplement 1*) were anatomically defined (Materials and methods). Here, MRS data after extensive quality control (31/36 in hMT+, and 28/36 in EVC [primarily in V1]) were taken for further analysis (Materials and methods).

## GABA and Glu concentrations in hMT+ and EVC (primarily in V1) and their relation to SI and BDT

An example of an MRS voxel located in hMT+ is shown in *Figure 3a*. LCModel fittings for GABA spectra from all subjects in hMT+ (*n*=31) and EVC (primarily in V1) (*n*=28) are illustrated in *Figure 3b* (color scale presents the BDT scores). We discerned a significant association between the inter-subjects' BDT scores and the GABA levels in hMT+ voxels, but not in EVC (primarily in V1) voxels. Quantitative analysis displayed that BDT significantly correlates with GABA concentrations in hMT+ voxels (*r*=0.39, p=0.03, *n*=31, *Figure 3c*). After using partial correlation to control for the effect of age, the relationship remains significant ($r_{partial}$ = 0.426, p=0.02, one participant excluded due to the age greater than mean + 2.5 SD). In contrast, there was no obvious correlation between BDT and GABA levels in EVC (primarily in V1) voxels (*Figure 3—figure supplement 1a*). We show that SI significantly correlates with GABA levels in hMT+ voxels (*r*=0.44, p=0.01, *n*=31, *Figure 3d*). In contrast, no significant correlation between SI and GABA concentrations in EVC (primarily in V1) voxels was observed (*Figure 3—figure supplement 1b*). These findings suggest that the relationship between motion suppression and GABA+ is specific to hMT+, but not in EVC (primarily in V1), which is in line with prior results (*Schallmo et al., 2018*). LCModel fittings for Glu spectra from all subjects in hMT+ (*n*=31) and EVC (primarily in V1) (*n*=28) voxels are presented in *Figure 3—figure supplement 2a*.

Unlike in the case of GABA, no significant correlations between BDT and Glu levels were found in both hMT+ and EVC (primarily in V1) voxels (*Figure 3—figure supplement 2b and c*). While, as expected (*Song et al., 2021*), we observed significant positive correlations between GABA and Glu concentrations in both hMT+ (*r*=0.62, p=0.0002, *n*=31) and EVC (primarily in V1) voxels (*r*=0.56, p=0.002, *n*=28; *Figure 3—figure supplement 3a and b*). Additionally, significant correlations between SI and BDT, duration threshold of small grating and BDT was discerned (*r*=0.59, p=0.0002, *n*=34, *Figure 3e*, $r_{partial}$ = 0.67, p<0.001, one participant excluded due to the age greater than mean + 2.5 SD; *r*=−0.43, p=0.016, $r_{partial}$ = 0.44, p=0.014, *Figure 3f*). While there was no significant correlation between duration threshold of large grating and BDT (*Figure 3g*), corroborating previous conclusions (*Melnick et al., 2013*). Two outliers evident in *Figure 3e* were excluded, with consistent results depicted in *Figure 3—figure supplement 4a*. Further, two outliers evident in *Figure 3d* were excluded, with consistent results depicted in *Figure 3—figure supplement 4b* .

## MT-frontal FC relates to SI and BDT

We next took the left hMT+ as the seed region and separately measured interregional FCs between the seed region and each voxel in the frontal regions (a priori search space). These measurements were correlated with performance in 3D visuo-spatial ability (BDT) to identify FCs with significant correlations. Results from connectivity-BDT analysis are summarized in *Table 1* and shown in *Figure 4a*. We found that brain regions with FC strength to the seed region (left hMT+) significantly correlated with BDT scores were situated within the canonical cognitive cores of fronto-parietal network (FPN) (Brodmann areas [BAs] 6, 9, 10, 46, 47; *Assem et al., 2020*; *Deary et al., 2010*; *Duncan et al., 2020*; *Duncan et al., 2000*; *Gray et al., 2003*; *Jung and Haier, 2007*). Across the whole-brain search, the similar FCs (between hMT+ and frontal cognitive cores) still showed significant correlations with BDT scores (*Supplementary file 1*; also shown in *Figure 4—figure supplement 1a*). Additionally, we identified certain parietal regions (BAs 7, 39, 40) with significant correlations between their connectivity to

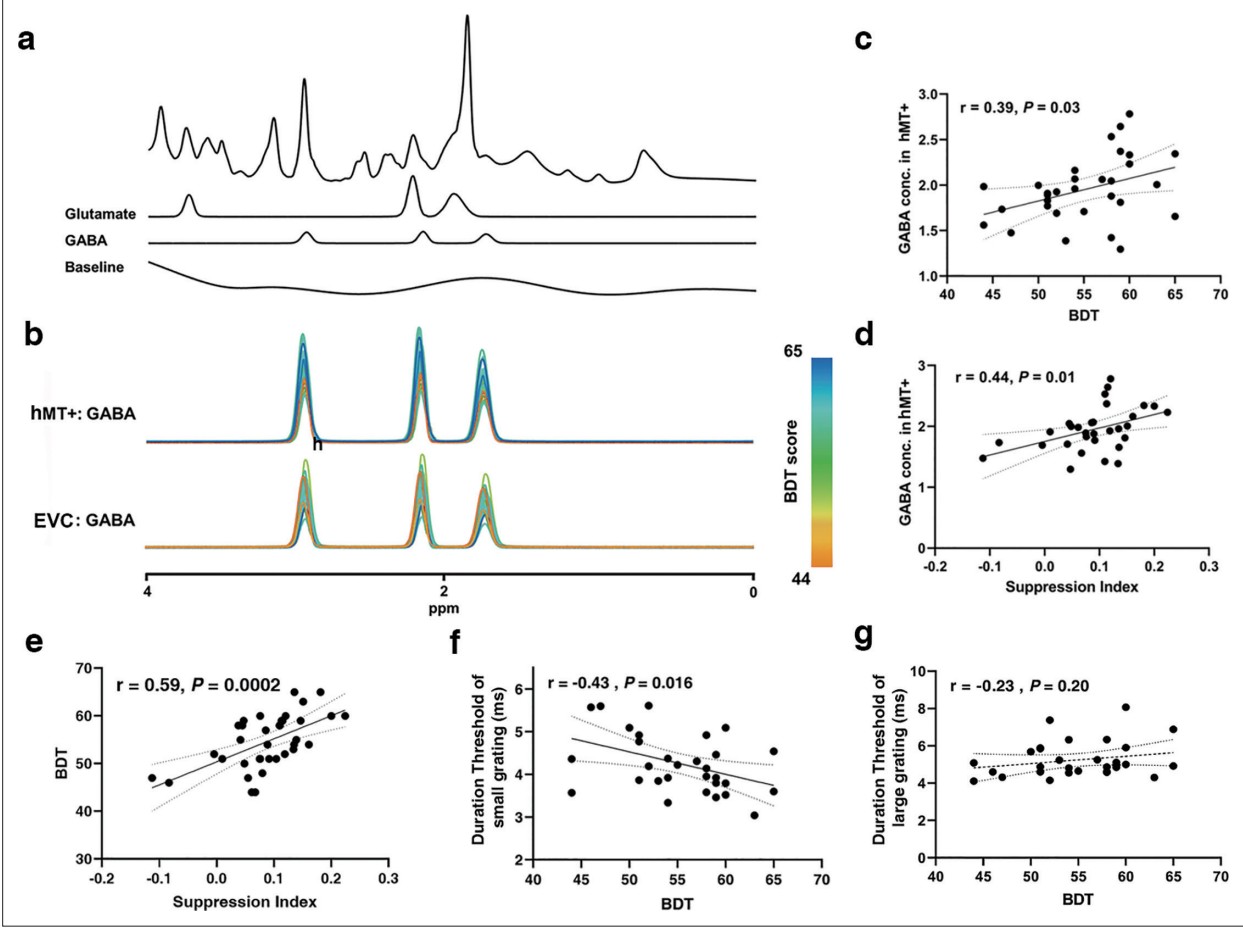

**Figure 3.** Magnetic resonance spectroscopy (MRS) spectra and the relationships between GABA levels and suppression index (SI)/block design test (BDT). (**a**) Example spectrum from the hMT+ voxel of one participant. The first line is the LCModel fitting result of all metabolites, and the following lines show the Glu and GABA spectra fitting with LCModel, and then the baseline. (**b**) Individual participants fitted GABA MRS spectra from the hMT+ (top) and early visual cortex (EVC) (primarily in V1) (bottom) voxels from baseline measurement. The colors of the GABA spectra represent the individual differences of BDT. The color bar represents the scores of BDT. (**c , d**) Pearson's correlations showing significant positive correlations between hMT+ GABA and BDT scores (**c**), between hMT+ GABA and SI (**d**). (**e**) Peason's correlation showing significant positive correlations between SI and BDT. (**f**) Peason's correlation showing significant negative correlations between BDT and duration threshold of small grating. (**g**) No correlation between BDT and duration threshold of large grating. The ribbon between dotted lines represents the 95% confidence interval, and the black regression line represents the Pearson's correlation coefficient (*r*). GABA and Glu concentrations (Conc.) are absolute, with units of mmol per kg wet weight (Materials and methods).

The online version of this article includes the following figure supplement(s) for figure 3:

**Figure supplement 1.** Relationships between GABA concentration in early visual cortex (EVC)1 (primarily in V1) and block design test (BDT)/suppression index (SI).

**Figure supplement 2.** Individual Glu magnetic resonance spectroscopy (MRS) spectra from hMT+/early visual cortex (EVC) (primarily in V1) regions and relationships between block design task (BDT) and Glu concentrations in hMT+/EVC (primarily in V1) regions.

**Figure supplement 3.** Correlations between GABA and Glu concentrations in hMT+ and early visual cortex (EVC) (primarily in V1) regions.

**Figure supplement 4.** Two linear correlations show without outlines results.

the left hMT+ and the BDT scores (*Supplementary file 1*; also shown in *Figure 4—figure supplement 1a*). These significant connections between hMT+ and FPN system suggest that left hMT+ is involved in the efficient information integration network mediating the visuo-spatial component of gF.

To address the question whether spatial suppression plays a role, we correlated hMT+-based global FCs with SI. Though spatial suppression during motion perception (quantified by SI) is considered to be the function of area hMT+(*Gautama and Van Hulle, 2001*; *Tadin et al., 2011*), the top-down modulation from the frontal cortex can increase surround suppression (*Liu et al., 2016*). Our

**Table 1.** Functional connectivity (FC) of voxels showing significant correlation with block design test (BDT) scores across subjects in frontal cortex.

| FC number | Connected regions | BA | Size | Peak coordinate MNI (*x, y, z*) | r | p |
|---|---|---|---|---|---|---|
| 1 | Frontal_Sup_Orb_R | 11 | 33 | (12,63,–19.5) | –0.57 | 0.0011 |
| 2 | Frontal_Inf_Orb_L | 47 | 24 | (–34.5,28.5,–13.5) | –0.63 | 0.0003 |
| 3 | Frontal_Med_Orb_R | 11 | 41 | (3,43.5,–12) | –0.58 | 0.0009 |
| 4 | Frontal_Inf_Orb_R | 47 | 48 | (–31.5,24,–12) | 0.59 | 0.0008 |
| 5 | Frontal_Inf_Orb_R | 47 | 29 | (25.5,30,–13.5) | 0.67 | 0.0001 |
| 6 | Insula_L | \ | 26 | (–28.5,27,0) | 0.67 | 0.0001 |
| 7 | Frontal_Inf_Oper_R | 45 | 41 | (43.5,16.5,6) | 0.64 | 0.0002 |
| 8 | Frontal_Sup_R | 10 | 25 | (31.5,57,9) | 0.59 | 0.0008 |
| 9 | Frontal_Mid_L | 10 | 82 | (–33,48,12) | 0.62 | 0.0003 |
| 10 | Frontal_Inf_Oper_R | 44 | 49 | (51,7.5,21) | 0.59 | 0.0007 |
| 11 | Frontal_Inf_Oper_R | 46 | 96 | (49.5,16.5,28.5) | –0.62 | 0.0003 |
| 12 | Frontal_Mid_L | 10 | 32 | (–31.5,49.5,24) | 0.57 | 0.0012 |
| 13 | Frontal_Mid_R | 10 | 102 | (31.5,36,30) | 0.59 | 0.0009 |
| 14 | Precentral_L | 6 | 46 | (–49.5,–1.5,34.5) | 0.59 | 0.0007 |
| 15 | Frontal_Mid_R | 9 | 107 | (51,19.5,40.5) | –0.67 | 0.0001 |
| 16 | Frontal_Sup_L | 9 | 35 | (–9,60,37.5) | –0.69 | 0.0001 |
| 17 | Frontal_Sup_Medial_R | 9 | 74 | (4.5,52.5,43.5) | –0.57 | 0.0011 |
| 18 | Frontal_Sup_R | 6 | 136 | (28.5,–7.5,63) | –0.64 | 0.0002 |
| 19 | Supp_Motor_Area_L | 6 | 48 | (–10.5,6,54) | 0.63 | 0.0003 |
| 20 | Frontal_Mid_L | 6 | 119 | (–24,4.5,55.5) | 0.63 | 0.0003 |
| 21 | Precentral_L | 6 | 229 | (–24,–18,66) | 0.60 | 0.0005 |
| 22 | Frontal_Sup_R | 6 | 32 | (16.5,–18,67.5) | 0.68 | 0.0001 |
| 23 | Precentral_R | 6 | 80 | (30,–24,70.5) | 0.70 | 0.0001 |
| 24 | Precentral_R | 6 | 23 | (16.5,–25.5,76.5) | 0.70 | 0.0001 |

Single voxel threshold $p<0.005$ (*t*>3.057 or *t*<–3.057), adjacent size ≥23 voxels (AlphaSim corrected).

FC-SI analysis in the frontal regions (a priori search space) displayed three brain regions in which FCs strength significantly correlated with SI: right BA4/6, left BA6, and right BA46 (summarized in *Supplementary file 2*, and shown in *Figure 4b*). Across the whole-brain search, we identified total seven brain regions in which FCs strength significantly correlated with SI, and three of these were in the frontal cortex. This is consistent with the results obtained by the FC-SI analysis in a priori search space (frontal cortex; *Supplementary file 3* and *Figure 4—figure supplement 1b*).

We also did the V1 FC-BDT correlations as control analysis (*Figure 4—figure supplement 2*). Only positive correlations in the frontal area were detected, suggesting that in the 3D visuo-spatial intelligence task, V1 plays a role in feedforward information processing. However, hMT+, which showed specific negative correlations in the frontal, be suggested involving in the inhibition mechanism. These results further emphasize the de-redundancy function of hMT+ in 3D visuo-spatial intelligence.

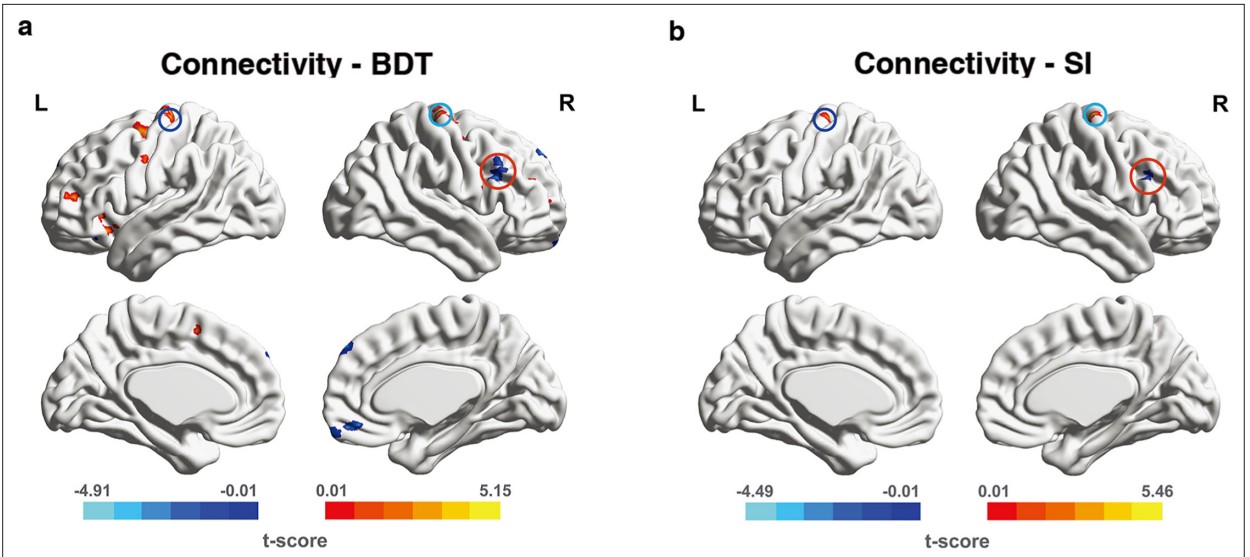

**Figure 4.** Significant functional connectivities (FCs) from connectivity-behavior analyses in a priori search space. The seed region is the left hMT+. The significant FCs are obtained from a priori space (frontal cortex). (**a**) The significant FCs obtained from connectivity-BDT analysis. Single voxel threshold p<0.005, adjacent size ≥ 23 (*AlphaSim* correcting, Materials and methods). (**b**) The significant FCs obtained from connectivity-SI analysis. Single voxel threshold p<0.005, adjacent size ≥ 22 (*AlphaSim* correcting, Materials and methods). Positive correlations are shown in warm colors, while negative correlations are shown in cold colors. The paired deep blue, light blue, red circles on (**a**) and (**b**) indicate the overlap regions in left BA6, right BA6, right BA46(DLPFC) between connectivity-BDT analysis and connectivity-SI analysis, respectively.

The online version of this article includes the following figure supplement(s) for figure 4:

**Figure supplement 1.** Significant functional connectivities (FCs) searched from connectivity-behavior (block design test [BDT]/surround suppression [SI]) analyses in the whole brain.

**Figure supplement 2.** Significant functional connectivities (FCs) searched from connectivity-behavior (block design test [BDT]) analyses in the frontal and whole brain.

## Local hMT+ GABA acts on SI and BDT via global hMT-frontal connectivity

To determine whether local neurotransmitter levels (such as GABA and Glu) in the hMT+ region mediate the broader 3D visuo-spatial ability of BDT, which as a component of gF, is linked to the frontal cortex (*Fangmeier et al., 2006*), we correlated the significant FCs of hMT-frontal in *Figure 4a* (also shown in *Table 1*) with the GABA and Glu levels in hMT+ region. The results revealed that only two FCs significantly correlated with inhibitory GABA levels in hMT+: (1) the FC of left hMT+-right BA46 (significantly negative correlation, $r = –0.56$, p=0.02, $n=29$, false discovery rate [FDR] correction, *Figure 5a* left); (2) the FC of left hMT+-right BA6 (significantly positive correlation, $r=0.69$, p=0.002, $n=29$, FDR correction, *Figure 5b* left; also shown in *Table 2*). There were no significant correlations between these FCs and the excitatory Glu levels in hMT+ (*Table 2*). Across the whole-brain search, we obtained the same two hMT+-frontal FCs significantly correlating with both hMT+ GABA levels and BDT (*Supplementary file 4*), this is consistent with the results in a priori search space (frontal cortex; *Table 2*). We then correlated the significant FCs in *Figure 5b* (also in *Supplementary file 2*) with GABA and Glu concentrations in hMT+ and found that almost all the correlations are significant except one (between the FC of left hMT+-right BA46 and the Glu levels in hMT+; *Supplementary file 5*). Among the three FCs, the clusters of two FCs have substantial voxel overlap with the FCs we found by the connectivity-BDT analysis (*Figure 5a and b*). Across the whole-brain search, there were total seven brain regions in which FCs strength were significantly correlated with SI, all the seven FCs significantly correlated the hMT+ GABA levels, while no FC had significant correlation with the hMT+ Glu levels (*Supplementary file 6*).

Taken together, our results displayed that the overlap FCs from the analyses of connectivity-behavior (BDT and SI) -GABA are the hMT+-BA46 and hMT+-BA6 (*Figure 5a and b*). These results suggest that the FCs of hMT+-frontal regions (BA46 and BA6) coupling with local hMT+ GABA provides the neural

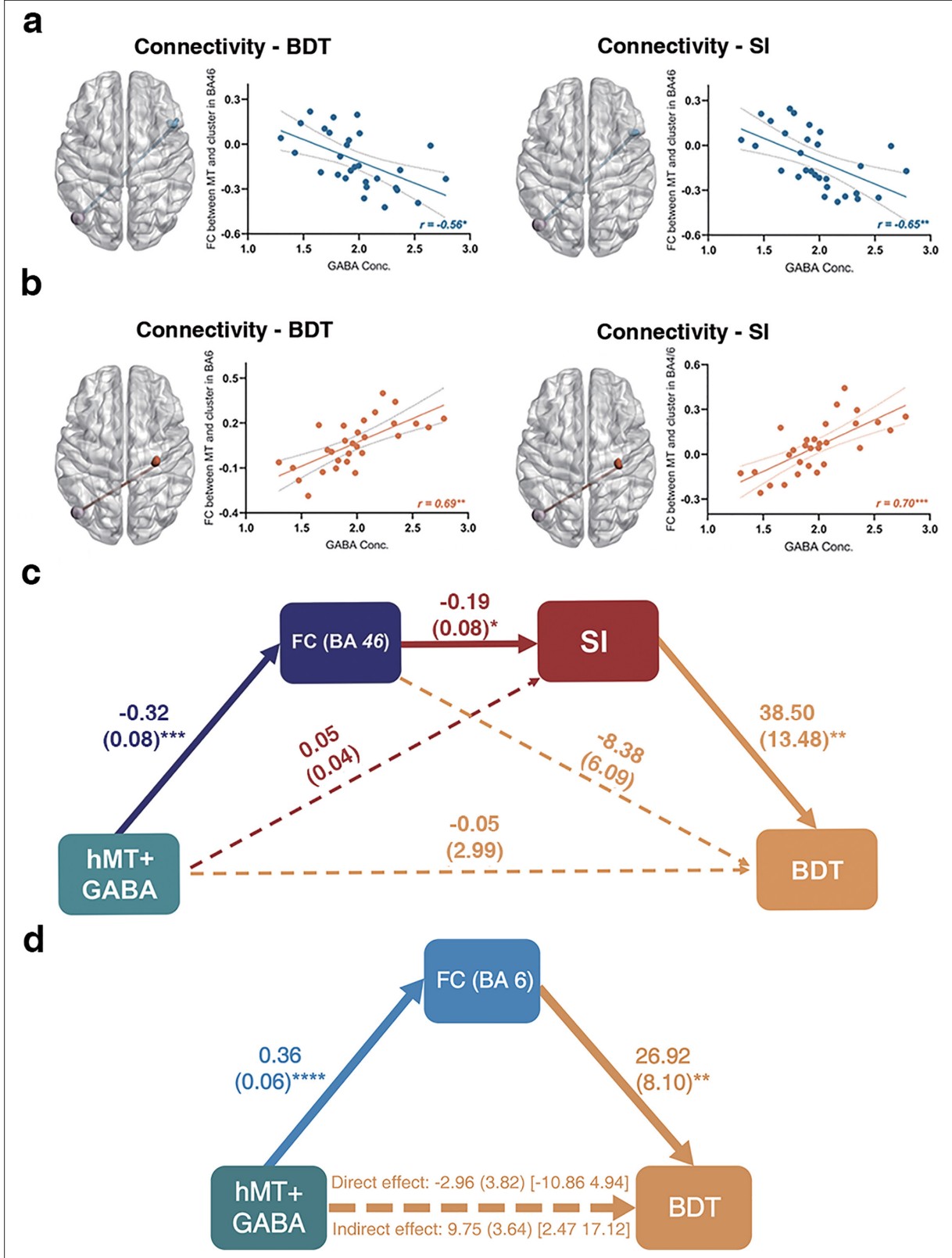

**Figure 5.** Local hMT+ GABA acts on suppression index (SI) and block design test (BDT) via global hMT-frontal connectivity. (**a**) Significant negative correlation between the functional connectivity (FC) of left hMT+-right DLPFC (BA46) and hMT+ GABA (false discovery rate [*FDR*] correction). (**b**) Significant positive correlation between the FC of left hMT+-right (pre) motor cortex (BA4/6) and hMT+ GABA (*FDR* correction). In (**a**) and (**b**), left: the significant FCs obtained from connectivity-BDT analysis; right: the significant FCs obtained from connectivity-SI analysis. (**c**) Significant pathways: hMT+

*Figure 5 continued on next page*

*Figure 5 continued*

GABA→ FC (left hMT+-right BA46, negative correlation) → SI (negative correlation) → BDT (positive correlation). This pathway can explain 34% of the variance in BDT. (**d**) Significant pathways: hMT+ GABA → FC (left hMT+-right BA6, positive correlation) → BDT (positive correlation). The bolded lines represent the hypothesized mediation effect. The dotted lines represent alternative pathways. *: p<0.05; **: p<0.01; ***: p<0.001.

The online version of this article includes the following figure supplement(s) for figure 5:

**Figure supplement 1.** The alternative serial mediation models from local hMT+ GABA to global performance of block design test (BDT).

basis for both the simple motion perception (quantified by SI) and the complex 3D visuo-spatial ability (quantified by BDT).

In order to fully investigate the potential roles of the multiples variables contributing to BDT scores, serial mediation analyses (*Hayes, 2013*) were applied to both the MR and behavioral data. Following our hypothesis, the independent variable (*X*) is hMT+ GABA, the dependent variable (*Y*) is BDT scores, the covariate is the age, and the mediators are FC (M1) and SI (M2). We used the overlap clusters

**Table 2.** Correlations between functional connectivity (FC) in Table 1 and GABA/Glu concentrations in hMT+.

| FC number | hMT+ GABA concentrations | | | hMT+ Glu concentrations | | |
|---|---|---|---|---|---|---|
| | *r* | *p* | *FDR* | *r* | *p* | *FDR* |
| 1 | −0.07 | 0.72 | 0.75 | −0.11 | 0.58 | 0.85 |
| 2 | −0.28 | 0.14 | 0.36 | −0.27 | 0.15 | 0.81 |
| 3 | −0.13 | 0.52 | 0.59 | −0.07 | 0.71 | 0.85 |
| 4 | 0.10 | 0.59 | 0.64 | 0.12 | 0.54 | 0.85 |
| 5 | 0.14 | 0.48 | 0.58 | 0.24 | 0.21 | 0.81 |
| 6 | 0.31 | 0.11 | 0.33 | 0.15 | 0.43 | 0.85 |
| 7 | 0.20 | 0.30 | 0.48 | 0.07 | 0.74 | 0.85 |
| 8 | 0.14 | 0.45 | 0.58 | 0.05 | 0.79 | 0.85 |
| 9 | 0.20 | 0.30 | 0.48 | 0.10 | 0.60 | 0.85 |
| 10 | −0.13 | 0.49 | 0.58 | −0.16 | 0.41 | 0.85 |
| 11 | **−0.56** | **0.0018** | **0.02*** | −0.22 | 0.25 | 0.81 |
| 12 | 0.18 | 0.34 | 0.51 | 0.15 | 0.43 | 0.85 |
| 13 | 0.20 | 0.30 | 0.48 | 0.05 | 0.81 | 0.85 |
| 14 | 0.39 | 0.04 | 0.14 | 0.22 | 0.24 | 0.81 |
| 15 | −0.40 | 0.03 | 0.12 | −0.21 | 0.27 | 0.81 |
| 16 | −0.27 | 0.15 | 0.36 | −0.12 | 0.53 | 0.85 |
| 17 | 0.17 | 0.37 | 0.52 | 0.06 | 0.74 | 0.85 |
| 18 | 0.26 | 0.18 | 0.39 | 0.16 | 0.40 | 0.85 |
| 19 | 0.39 | 0.03 | 0.12 | 0.31 | 0.10 | 0.81 |
| 20 | 0.01 | 0.98 | 0.98 | 0.14 | 0.46 | 0.85 |
| 21 | 0.40 | 0.03 | 0.12 | 0.24 | 0.21 | 0.81 |
| 22 | 0.22 | 0.25 | 0.48 | 0.06 | 0.76 | 0.85 |
| 23 | **0.69** | **0.0001** | **0.002**** | 0.47 | 0.01 | 0.24 |
| 24 | 0.41 | 0.03 | 0.12 | 0.001 | 0.97 | 0.97 |

*: $p_{FDR} < 0.05$; **: $p_{FDR} < 0.01$; ***: $p_{FDR} < 0.001$; Bold font indicates the significant correlations survived from multi-correlation correction.

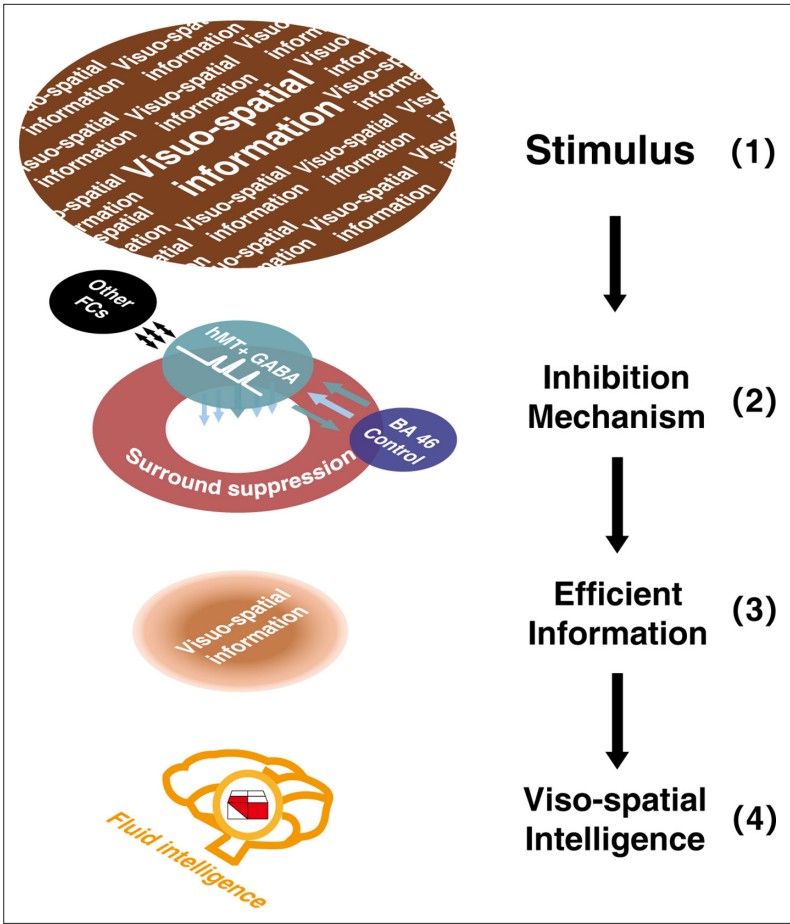

**Figure 6.** Sketch depicting the multi-level inhibitory mechanisms centered on hMT+ GABA contributing to visuo-spatial intelligence. Inhibitory GABA in hMT+ (a sensory cortex, shown in green circle), coupling with the functional connectivity between hMT+ and BA46 (cognitive control core, shown in purple circle), and mediated by motion surround suppression (shown in red circle), contributes to visuo-spatial intelligence (block design test [BDT], 3D domain, shown in red and white building blocks). In this sketch, the two-colored parallel arrows show the negative FC between hMT+ and BA46, the colored arrows below the green circle display the inhibition mechanisms centered on hMT+ GABA (2), filtered the irrelevant information in (1) and focused on the efficient visuo-spatial information (3). Black long arrows display the direction of information flow: from input information (1) to visuo-spatial intelligence (4).

The online version of this article includes the following figure supplement(s) for figure 6:

**Figure supplement 1.** Correlation matrix shows correlations between all measured in our biochemistry and behavior level.

from the analyses of connectivity-BDT-GABA and connectivity-SI-GABA to compute the FC of hMT+-BA46, one participant was excluded due to his age greater than mean + 2.5 SD. The serial mediation model is shown in *Figure 5c*. GABA levels in hMT+ significantly negatively correlated with the FC of hMT+-BA46 ($\beta$ = –0.32, p=0.0009), which in turn significantly negatively correlated with SI ($\beta$ = –0.19, p=0.035), and consequently, significantly positively correlated with BDT ($\beta$=38.5, p=0.009). Critically, bootstrapped analyses revealed that our hypothesized indirect effect (i.e. hMT+ GABA → FC of hMT+-BA46 → SI → BDT) was significant ($\beta$=2.28, SE = 1.54, 95% CI = [0.03, 5.94]). The model accounted for 34% of the variance in BDT. However, when considering the hMT-BA6 FC as the mediator M1, the serial model does not show a significant indirect effect. Consequently, we explored a mediation model, which revealed that the hMT-BA6 FC totally mediates the relationship between GABA and BDS (*Figure 5d*). For sensitivity purposes, we tested the alternative models, in which the order of the mediators was reversed. The pathway that hMT+ GABA was predicted to be associated

with SI, followed by the FC of hMT+-BA46, and then BDT, did not yield the chained mediation effects on BDT (**Figure 5—figure supplement 1**).

To summarize (shown in **Figure 6**), the results from the serial mediation analyses are consistent with our hypothesis. That is, higher GABAergic inhibition in hMT+ relates to stronger negative FC between hMT+ and BA46, leading to enhanced ability for surround suppression (filtering out irrelevant information; **Tadin, 2015**), which ultimately resulting in more efficient visual 3D processing as key component of gF (the higher BDT scores).

## Discussion

Here, we provide evidence that hMT+ inhibitory mechanisms mediate processing in the visuo-spatial component (3D domain) of gF on multiple levels, i.e., from molecular over brain connectivity to behavior. First, this study found that higher hMT+ inhibitory GABA levels (but not excitatory Glu) relate to FC between hMT+ and BA46 that contribute to both SI and BDT. Our serial mediation analyses indicate that the inhibitory mechanisms related to hMT+ and its GABA levels in hMT+ (but not Glu), FCs of hMT+-BA46 coupling with hMT+ inhibitory GABA (but not excitatory Glu), and behavior (SI indexing perceptual suppression in hMT+) predict the inter-subject variance in the 3D gF task (BDT; **Figure 5c**). Second, we demonstrate discrete GABAergic inhibition mechanisms in hMT+ that mediate the strong FCs between hMT+-frontal regions (BA46 and BA6): significant negative correlation with the FC of hMT+-BA46 (**Figure 5a**), whereas there is significant positive correlation with the FC of hMT+-BA6 (**Figure 5b**). This indicates that different frontal regions, DLPFC (BA46) and premotor cortex (BA6), contribute uniquely to gF through hMT+-based inhibitory mechanisms.

The goal of our research is to reveal that the inhibitory (not excitatory) mechanism in hMT+ contributes to multi-level processing in 3D visuo-spatial ability (BDT). Monkey electrophysiological experiments revealed that selective attention gates the visual cortex, including area MT, effectively suppressing the irrelevant information (**Everling et al., 2002**; **Treue and Maunsell, 1996**). These findings align with the 'neural efficiency' hypothesis of intelligence (**Haier et al., 1988**), which puts forward the human brain's ability to suppress the repetition of information. Neural suppression is associated with the balance between excitation and inhibition (EIB), usually represented by covariation between Glu and GABA (**Ozeki et al., 2009**). Here, this study exploited the high spectral resolution afforded by ultra-high field (7T) MRS to reliably resolve GABA measurement, to adequately discriminate the Glu and glutamine signals, and to resolve the high-accuracy Glu measurement (**Ende, 2015**).

This work implemented the MRS scanning in hMT+ (3D visual domain) and EVC (primarily in V1) (2D visual domain) regions and found that hMT+ inhibitory GABA (but not excitatory Glu) significantly correlated with BDT, i.e., the higher GABA levels in hMT+ (rather than excitatory Glu) relate to higher visual 3D processing (BDT; **Figure 3c**). Basically, this study contains the data of SI, BDT, GABA in MT+ and EVC (primarily in V1), Glu in MT+, and EVC (primarily in V1)-all six measurements. We made a correlation matrix to reporting all values in **Figure 6—figure supplement 1**.

We searched the global hMT+-based FCs with the connectivity-BDT analyses (in a priori search space and whole-brain search to valid), and then, correlated these significant FCs with the GABA and Glu concentrations in hMT+. We found two FCs (hMT+-BA46 and hMT+-BA6) significantly correlating with hMT+ inhibitory GABA (whereas no FC significantly correlated with hMT+ excitatory Glu). Accordingly, our results emphasize the importance of hMT+ inhibitory GABA (but not excitatory Glu) in processing the 3D visual-spatial intelligence (BDT).

Our recent human study (**Song et al., 2021**) and other study's animal experiments **Ozeki et al., 2009**; **Sato et al., 2016** demonstrated that the conjoint action of inhibition (GABA) and excitation (Glu) underlies visual spatial suppression. In this work, our novel data show the chained mediation effects from local hMT+ GABA to more global BDT: hMT+ GABA → FC (hMT+ and BA46) → SI → BDT. Thereby, our data indicate that inhibitory mechanisms in hMT+, from the biochemical level of GABA over FC to the behavioral level, can predict the inter-subject variance in the 3D gF task (BDT; **Figure 5c**).

Another interesting finding reveals that GABAergic inhibition in hMT+ coupling with distinct FC patterns between BA46-hMT+ and BA6-hMT+. A previous human fMRI experiment found that the positive and negative correlations between BDT and the activation of frontal regions appeared at different reasoning phases (validation or integration phases during reasoning; **Fangmeier et al., 2006**). On the one hand, a monkey electrophysiological experiment reported the delayed modulation

from PFC (especially in DLPFC; BA46) to area MT during a visual motion task (*Zaksas and Pasternak, 2006*). Computational models converged with empirical data of awake monkey experiments slowing temporal modulation from PFC to MT/medial superior temporal (MST; *Donner et al., 2009*; *Siegel et al., 2015*; *Wang, 2002*; *Wimmer et al., 2015*). On the other hand, human MEG studies (*Donner et al., 2009*; *Wilming et al., 2020*) reported that the gamma-band activity in the visual cortex (including area MT) exhibited high coherence with the activity in (pre-) motor regions (BA4/6). These results suggest that the relation of long-range FC and local inhibitory mechanism (hMT+ GABA) support our findings that inhibition in hMT+ contributes to efficient long-range integration and coordination in distant brain areas like the prefrontal and premotor cortex.

How does hMT+ assemble into the cognitive system as an intellectual hub rather than a simple input module? The results in *Figure 5a and b* showed that the overlap brain regions from the analyses of connectivity-BDT-GABA/connectivity-SI-GABA are the hMT+-BA46. This overlap couples with local visual suppression (SI) and consequently plays an important role in intelligence (BDT). The direction discrimination task in this work (the visual motion paradigm of center-surround antagonism) was previously considered a mainly local function of hMT+ (*Melnick et al., 2013*; *Tadin, 2015*; *Tadin et al., 2003*). However, our results with connectivity-SI analyses revealed that both local (FC within BA18) and global brain connectivity (FC between hMT+ and frontal regions) contribute to SI (*Supplementary file 3*). In human psychophysical experiments (*Melnick et al., 2013*; *Tadin et al., 2003*) the brief stimulus duration (~100 ms) in motion discrimination precludes most top-down attentional effects (*Wang, 2002*; *Zaksas and Pasternak, 2006*), while attention, which predicted the performance of the motion discrimination task, was sustained throughout the stimulus intervals (*Siegel et al., 2015*). Furthermore, animal experiments have revealed that the local circuits in the visual cortex combining with top-down modulation and intracortical horizontal connection mediate the visual-spatial suppression (*Angelucci et al., 2002*; *Keller et al., 2020*; *Li et al., 2019*; *Zhang et al., 2014*).

Our results (shown in *Figure 5a and b*, right) present the intrinsic binding of local GABAergic inhibition in hMT+, which suppresses redundancy of visual motion processing (SI), and the activity of brain connectivity between hMT+ and frontal regions. These individual inherent traits may contribute to the individual difference in 3D visuo-spatial ability (*Figure 5a and b*, left). A candidate divisive normalization model (*Carandini and Heeger, 2011*; *Reynolds and Heeger, 2009*) can explain how such reverberation affects the process of suppressing the irrelevant information, from perception to intelligence (*Melnick et al., 2013*; *Tadin, 2015*). We summarize a framework (*Figure 6*) to indicate and visualize our findings.

Recently, Duncan et al. demonstrated coding of gF in distributed regions, defining them as part of multi-demand (MD) systems (*Assem et al., 2020*; *Duncan et al., 2020*). The MD system encompasses a range of cognitive domains, including working memory, mathematics, language, and relational reasoning. According to *Melnick et al., 2013*, motion surround suppression (SI) and time thresholds for small and large gratings, which reflect hMT+ functionality, are correlated with Verbal Comprehension, Perceptual Reasoning, Working Memory, and Processing Speed indicators. Additionally, Fedorenko et al. identified MD activation regions around the occipito-temporal areas, potentially overlapping with hMT+ (*Fedorenko et al., 2013*). As a key region in the representation of sensory flows (including optic and auditory flows; *Fetsch et al., 2011*; *Gu et al., 2006*), hMT+ shows potential to be central to the MD system. Future research could focus on multi-task paradigms to further investigate the mechanisms of hMT+ and its relationship with broader cognitive functions.

Together, this study offers a comprehensive insight into how the information exchange and integration between the sensory cortex (hMT+) and cognition core of BA46, coupling with the hMT+ GABA, can predict the performance of 3D visuo-spatial ability (BDT). Our results provide direct evidence that a sensory cortex area (hMT+), its GABA biochemistry, FC, and cognition behavior levels can assemble into complex cognition as an intellectual hub.

## Materials and methods
### Subjects
Thirty-six healthy subjects (18 female, mean age: 23.6 years±2.1, range: 20–29 years) participated in this study, they were recruited from Zhejiang University. All subjects had normal or corrected-to-normal vision. In addition, they reported no psychotropic medication use, no illicit drug use within

the past month, no alcohol use within 3 days prior to scanning, and right-handed. This experiment was approved by the Ethics Review Committee of Zhejiang University and conducted in accordance with the Helsinki Declaration. All participants signed informed consent forms prior to the start of the study and were compensated for their time. All subjects participated in the motion spatial suppression psychophysical, resting-state fMRI, and MRS (hMT+ and EVC [primarily in V1] regions, in random sequence) experiments, but only part of the MRS data (31/36 in hMT+ region and 28/36 in EVC [primarily in V1] region) survived quality control (see the part of MRS data processing). The sample size is determined by the statistic requirement (30 sample for Pearson's correlation statistical analysis).

## Motion surrounding suppression measurement

All stimuli were generated using MATLAB (MathWorks, Natick, MA, USA) with Psychophysics Toolbox (*Brainard, 1997*), and were shown on a linearized monitor (1920×1080 resolution, 100 Hz refresh rate, Cambridge Research System, Kent, UK). The viewing distance was 72 cm from the screen, with the head stabilized by a chinrest. Stimuli were drawn against a gray (56 cd per m$^{-2}$) background.

A schematic of the stimuli and trial sequences is shown in our recent study (*Song et al., 2021*). The stimulus was a vertical drifting sinusoidal grating (contrast, 50%; spatial frequency, 1 cycle/°; speed, 4°/s) of either small (diameter of 2°) or large (diameter of 10°) size. The edge of the grating was blurred with a raised cosine function (width, 0.3°). A cross was presented in the center of the screen at the beginning of each trial for 500 ms, and participants were instructed to fixate at the cross and to keep fixating at the cross throughout the trial. In each trial, a grating of either large or small size was randomly presented at the center of the screen. The grating drifted either leftward or rightward, and participants were asked to judge the perceived moving direction by a key press. Response time was not limited. The grating was ramped on and off with a Gaussian temporal envelope, and the grating duration was defined as 1 SD of the Gaussian function. The duration was adaptively adjusted in each trial, and duration thresholds were estimated by a staircase procedure. Thresholds for large and small gratings were obtained from a 160-trial block that contained four interleaved 3-down/1-up staircases. For each participant, we computed the correct rate for different stimulus durations separately for each stimulus size. These values were then fitted to a cumulative Gaussian function, and the duration threshold corresponding to the 75% correct point on the psychometric function was estimated for each stimulus size.

Stimulus demonstration and practice trials were presented before the first run. Auditory feedback was provided for each wrong response. To quantify the spatial suppression strength, we calculated the spatial SI, defined as the difference of $\log_{10}$ thresholds for large versus small stimuli (*Schallmo et al., 2018*; *Tadin et al., 2003*):

$$SI = \log_{10}\left(large\ threshold\right) - \log_{10}\left(small\ threshold\right) \qquad (1)$$

## Block design task measurement

The block design task was administered in accordance with the WAIS-IV manual (*Wechsler, 2008*). Specifically, participants were asked to rebuild the figural pattern within a specified time limit using a set of red and white blocks. The time limits were set as 30–120 s according to different levels of difficulty. The patterns were presented in ascending order of difficulty, and the test stopped if two consecutive patterns were not constructed in the allotted time. The score was determined by the accomplishment of the pattern and the time taken. A time bonus was awarded for rapid performance in the last six patterns. The score ranges between 0 and 66 points, with higher scores indicating better perceptual reasoning.

## MR experimental procedure

MR experiments were performed in a 7T whole-body MR system (Siemens Healthcare, Erlangen, Germany) with a Nova Medical 32-channel array head coil. Sessions included resting-state functional MRI, fMRI localizer scan, structural image scanning, and MRS scan. Resting-state scans were acquired with 1.5 mm isotropic resolution (transverse orientation, TR/TE = 2000/20.6 ms, 160 volumes, slice number = 90, flip angle = 70°, eyes closed). Structural images were acquired using an MP2RAGE sequence (TR/TI1/TI2=5000/901/3200 ms) with 0.7 mm isotropic resolution. MRS data were collected within two regions (hMT+ and EVC [primarily in V1]) for each subject, and we divided them into

two sessions to avoid discomfort caused by long scanning. The order of MRS VOIs (hMT+ and EVC [primarily in V1]) in the two sessions was counterbalanced across participants. Interval between two sessions was used for block design and motion discrimination tasks. One session included fMRI localizer scan, structural image scanning, and MRS scan for the hMT+ region; the other session included structural image scan, and MRS scan for the EVC (primarily in V1) region. Spectroscopy data were acquired using a $^1$H-MRS single-voxel short-TE STEAM (Stimulated Echo Acquisition Mode) sequence (*Frahm et al., 1989*; TE/TM/TR = 6/32/7100 ms) with 4096 sampling points, 4 kHz bandwidth, 16 averages, 8 repetitions, 20×20×20 mm$^3$ VOI size, and VAPOR (variable power and optimized relaxation delays) water suppression (*Tkác et al., 1999*). Prior to acquisition, first- and second-order shims were adjusted using FASTMAP (fast, automatic shimming technique by mapping along projections; *Gruetter, 1993*). Two non-suppressed water spectra were also acquired: one for phase and eddy current correction (only RF pulse, 4 averages) and another for metabolite quantification (VAPOR none, 4 averages). Voxels were positioned based on anatomical landmarks using a structural image scan collected in the same session, while avoiding contamination by CSF, bone, and fat. The hMT+ VOIs were placed in the ventrolateral occipital lobe, which was based on anatomical landmarks (*Dumoulin et al., 2000*; *Schallmo et al., 2018*). We did not distinguish between the MT and MST areas in these hMT+ VOIs (*Huk et al., 2002*). For 14 subjects, we also functionally identified hMT+ as a check on the placement of the VOI. A protocol was used with a drifting grating (15% contrast) alternated with a static grating across blocks (10 s block duration, 160 TRs total). Using fMRI BOLD signals, these localizer data were processed online to identify the hMT+ voxels in the lateral occipital cortex, which responded more strongly to moving vs. static gratings. In addition, we only used the left hMT+ as the target region to scan, which was motivated by studies showing that left hMT+ was more effective at causing perceptual effects (*Tadin et al., 2011*). For EVC (primarily in V1) region, the VOI was positioned on each subject's calcarine sulcus on the left side (*Tadin et al., 2011*) based on anatomical landmarks (*Boucard et al., 2007*; *Dumoulin et al., 2000*).

## MRS data processing

Spectroscopy data were preprocessed and quantified using magnetic resonance signal processing and analysis, https://www.cmrr.umn.edu/downloads/mrspa/, which runs under MATLAB and invokes the interface of the LCModel (version 6.3-1L; *Chen et al., 2019*). First, we used the non-suppressed water spectra to perform eddy current correction and frequency/phase correction. Second, we checked the quality of each FID (16 averages) visually and removed those with obviously poor quality. Third, the absolute concentrations of each metabolite were quantitatively estimated via the water scaling method. For partial-volume correction, the tissue water content was computed as follows *Ernst et al., 1993*:

$$Tissue\ water\ content = fgm * 0.78\ + fwm * 0.65\ + fcsf * 0.97 \tag{2}$$

where *fgm*, *fwm*, and *fcsf* were the GM/WM/CSF volume fraction in MRS VOI and we used FAST (fMRI's automated segmentation tool, part of the FSL toolbox; *Zhang et al., 2001*) to segment the three tissue compartments from the T1-weighted structural brain images. For water T2 correction, we set water T2 as 47 ms (*Marjańska et al., 2012*). Our concentrations were mM per kg wet weight. Furthermore, LCModel analysis was performed on all spectra within the chemical shift range of 0.2–4.0 ppm.

Poor spectral quality was established by a Cramer-Rao lower bound of more than 20% (*Cavassila et al., 2001*), and some data were excluded from further analysis. The details were described in our recent paper (*Song et al., 2021*).

## Resting-state fMRI data processing and analysis

Resting-state functional image was analyzed in the Data Processing and Analysis for Brain Imaging DPABI toolbox (*Yan et al., 2016*) based on SPM 12 (http://www.fil.ion.ucl.ac.uk/spm/). The preprocessing steps included discard of the first five volumes, slice timing, realignment to the 90th slice, coregistration of each subject's T1-weighted anatomical and functional images, segmentation of the anatomical images into six types of tissues using DARTEL, linear detrend, regressing nuisance variables (including realignment Friston 24-parameter, global signal, white matter and CSF signal; *Friston et al., 1996*), normalization to the standard Montreal Neurological Institute (MNI) space with the voxel

size of 1.5×1.5×1.5 mm$^3$ using DARTEL, spatial smoothing with a Gaussian kernel of 3 mm full-width-half-maximum, and band-pass filtering with standard frequency band (0.01–0.1 Hz). Spherical ROI with a radius of 6 mm was placed in left MT. The coordinate for left MT (−46, −72, −4, in MNI space) was obtained by our localizer fMRI experiment. We calculated the seed-to-voxel whole-brain FC map for each subject. All the FC values were Fisher-Z-transformed.

We did a similar connectivity-behavior analysis to a previous study (*Song et al., 2008*). First, we computed the Pearson's correlation coefficient between BDT scores and the FC values across subjects in a voxel-based way. Then, to evaluate the significance, we transformed the *r*-value into *t*-value ($t = \frac{\sqrt{df \cdot r}}{\sqrt{1-r^2}}$), where *df* denotes the degrees of freedom, and *r* is the Pearson's correlation coefficient between BDT scores and the FC values. Here, *df* was equal to 27. The brain regions in which the FC values to the seed region was significantly correlated with the BDT scores were obtained with a threshold of p<0.005 for regions of a priori ($|t_{(27)}| \geq 3.057$, and adjacent cluster size ≥23 voxels; AlphaSim corrected), and p<0.01 for whole-brain analyses ($|t_{(27)}| \geq 2.771$, and adjacent cluster size ≥37 voxels; AlphaSim corrected).

## Statistical analysis

PROCESS version 3.4, a toolbox in SPSS, was used to examine the mediation model. There are some prerequisites for mediation analysis: the independent variable should be a significant predictor of the mediator, and the mediator should be a significant predictor of the dependent variable.

SPSS 20 (IBM, USA) was used to conduct all the remaining statistical analysis in the study. We evaluated the correlation of variables (GABA, Glu, SI, BDT) using Pearson's correlation analysis. Differences or correlations were considered statistically significant if p<0.05. Significances with multiple comparisons were tested with *FDR* correction. The effect of age on intelligence was controlled for by using partial correlation in the correlation analysis and was taken as a covariate in the serial mediation model analysis.

## Acknowledgements

The authors thank Prof. Dost Ongur and Fei Du for guidance on the MRS data processing. We thank Zhejiang University 7T Brain Imaging Research Center, and thank Guohua Xu and Fen Yang for technical assistance. We thank Prof. Xinyi Lai for supporting MRI data acquisition. This work was supported by STI 2030 - Major Projects (2021ZD0200401 to XMS, 2022ZD0206000 to RB), the National Natural Science Foundation of China Grants (U1909205, 61876222, 32000761, 82222032), Humanities and Social Sciences Ministry of Education (20YJC880095, 18YJA190001), the Key R&D Program of Zhejiang (2022C03096 to XMS, 2022ZJJH02-06 to GC), the European Union's Horizon 2020 Framework Program for Research and Innovation under the Specific Grant Agreement No. 785907 (Human Brain Project SGA2 to GN), and the MOE Frontier Science Center for Brain Science & Brain-Machine Integration, Zhejiang University.

## Additional information

### Funding

| Funder | Grant reference number | Author |
| --- | --- | --- |
| STI 2030 - Major Projects | 2021ZD0200401 | Xue Mei Song |
| STI 2030 - Major Projects | 2022ZD0206000 | Ruiliang Bai |
| The National Natural Science Foundation of China | 61876222 | Yong-Chun Cai |
| The National Natural Science Foundation of China | 82222032 | Ruiliang Bai |

| Funder | Grant reference number | Author |
|---|---|---|
| The National Natural Science Foundation of China | U1909205 | Gang Chen |
| Horizon 2020 Framework Programme | 785907 | Georg Northoff |
| Humanities and Social Sciences Ministry of Education | 20YJC880095 | Tengfei Wang |
| Humanities and Social Sciences Ministry of Education | 18YJA190001 | Yong-Chun Cai |
| The Key R&D Program of Zhejiang | 2022C03096 | Xue Mei Song |
| The Key R&D Program of Zhejiang | 2022ZJJH02-06 | Gang Chen |
| The National Natural Science Foundation of China | 32000761 | Tengfei Wang |

The funders had no role in study design, data collection and interpretation, or the decision to submit the work for publication.

## Author contributions

Yuan Gao, Data curation, Formal analysis, Investigation, Visualization, Methodology, Writing – original draft; Yong-Chun Cai, Software, Methodology, Writing – original draft; Dong-Yu Liu, Juan Yu, Formal analysis, Methodology; Jue Wang, Bin Xu, Data curation; Ming Li, Formal analysis; Tengfei Wang, Formal analysis, Methodology, Writing – review and editing; Gang Chen, Writing – review and editing; Georg Northoff, Supervision, Writing – review and editing; Ruiliang Bai, Software, Formal analysis, Methodology, Writing – review and editing; Xue Mei Song, Conceptualization, Supervision, Funding acquisition, Writing – original draft, Project administration, Writing – review and editing

### Author ORCIDs

Tengfei Wang ⬡ https://orcid.org/0000-0002-1585-4143
Georg Northoff ⬡ https://orcid.org/0000-0002-5236-0951
Xue Mei Song ⬡ https://orcid.org/0000-0003-3624-4245

### Ethics

This experiment was approved by the Ethics Review Committee of Zhejiang University and conducted in accordance with the Helsinki Declaration. (Identifer:[2022]076). All participants signed informed consent forms prior to the start of the study and were compensated for their time.

Reviewer #1 (Public review): https://doi.org/10.7554/eLife.97545.4.sa1
Reviewer #3 (Public review): https://doi.org/10.7554/eLife.97545.4.sa2
Author response https://doi.org/10.7554/eLife.97545.4.sa3

# Additional files

## Supplementary files

• Supplementary file 1. Functional connectivity (FC) of voxels showing significant correlation with block design test (BDT) scores across subjects in whole brain.

• Supplementary file 2. Functional connectivities (FCs) of voxels showing significant correlation with suppression index (SI) across subjects in frontal cortex.

• Supplementary file 3. Functional connectivities (FCs) of voxels showing significant correlation with suppression index (SI) across subjects in whole brain.

• Supplementary file 4. Correlations between functional connectivity (FC) in *Supplementary file 1*

and GABA/Glu concentrations in hMT+.

• Supplementary file 5. Correlations between functional connectivities (FCs) in *Supplementary file 2* and GABA/Glu concentrations in hMT+.

• Supplementary file 6. Correlations between functional connectivities (FCs) in *Supplementary file 3* and GABA/Glu concentrations in hMT+.

• MDAR checklist

## Data availability

Source data are provided with this paper and have been archived at Zenodo.

The following dataset was generated:

| Author(s) | Year | Dataset title | Dataset URL | Database and Identifier |
|-----------|------|---------------|-------------|-------------------------|
| Yuan G | 2024 | GABA-ergic inhibition in human hMT+ predicts visuo-spatial intelligence mediated through the frontal cortex | https://doi.org/10.5281/zenodo.13753668 | Zenodo, 10.5281/zenodo.13753668 |

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
