## [Editor Report · eLife assessment]

This **important** study adopts a comprehensive approach: functional connectivity, biochemistry, and psychophysics to reveal a holistic understanding of the relationship between GABA-ergic inhibition in the human MT+ region and visuo-spatial intelligence. The evidence supporting the conclusion is **convincing**. The result advances our understanding of how the human MT+ is assemble into complex cognition as an intellectual hub, and will be of interest to researchers in psychology, cognitive science, and neuroscience.

---

## [Referee Report · Reviewer #1 (Public review)]

Summary:

The study of human intelligence has been the focus of cognitive neuroscience research, and finding some objective behavioral or neural indicators of intelligence has been an ongoing problem for scientists for many years. Melnick et al, 2013 found for the first time that the phenomenon of spatial suppression in motion perception predicts an individual's IQ score. This is because IQ is likely associated with the ability to suppress irrelevant information. In this study, a high-resolution MRS approach was used to test this theory. In this paper, the phenomenon of spatial suppression in motion perception was found to be correlated with the visuo-spatial subtest of gF, while both variables were also correlated with the GABA concentration of MT+ in the human brain. In addition, there was no significant relationship with the excitatory transmitter Glu. At the same time, SI was also associated with MT+ and several frontal cortex FCs.

Strengths:

(1) 7T high-resolution MRS is used

(2) This study combines the behavioral tests, MRS, and fMRI.

Major

I have no further comments. The approach and experiment are sound. The only overall drawback is the relatively low sample size.

Weaknesses:

(1) Line 138, "This finding supports the hypothesis that motion perception is associated with neural activity in MT+ area". This sentence is strange because it is a well-established finding in numerous human fMRI papers. I think the authors should be more specific about what this finding implies.

Response: We thank reviewer for pointing this out. We have revised it to:" This finding is in line with prior results, which indicates that motion perception is associated with neural activity in hMT+ area, but not in EVC (primarily in V1)" (lines 156-158)

Reply: This argument should be refined. Numerous studies have shown the key role of V1 in motion perception. V1 contains a vast proportion of direction selective neurons. I am asking how your results here are related to existing literature. This argument is incorrect and too rough. Can you please revise this?

(9) Line 213, as far as I know, the study (Melnick et al., 2013) is a psychophysical study and did not provide evidence that the spatial suppression effect is associated with MT+.

Response: We thank reviewer for pointing this out. It was a mistake to use this reference, and we have revised it accordingly. (line 242)

Reply: Thanks. New citation is good. But that paper is a modeling study. The direct empirical evidence on humans should be as follow:

Tadin, D., Silvanto, J., Pascual-Leone, A. & Battelli, L. (2011) Improved motion perception and impaired spatial suppression following disruption of cortical area MT/V5. Journal of Neuroscience, 31, 1279-1283.

---

## [Referee Report · Reviewer #3 (Public review)]

Summary:

This study aims to understand the role of GABA-ergic inhibition in the human MT+ region in predicting visuo-spatial intelligence through a combination of behavioral measures, fMRI (for functional connectivity measurement), and MRS (for GABA/glutamate concentration measurement). It provides useful evidence that GABA levels in the sensory cortex, such as in the human MT+, are associated with visuo-spatial ability, thus highlighting the importance of GABA-ergic inhibition in complex cognition.

Strengths:

(1) Comprehensive Approach: The study adopts a multi-level approach, i.e., neurochemical analysis of GABA levels, functional connectivity, and behavioral measures to provide a holistic understanding of the relationship between GABA-ergic inhibition and visuo-spatial intelligence.

(2) Sophisticated Techniques: The use of ultra-high field magnetic resonance spectroscopy (MRS) technology for measuring GABA and glutamate concentrations in the MT+ region is a recent development.

Weaknesses:

The authors have carefully addressed the major weaknesses previously mentioned.

---

## [Author Response]

The following is the authors’ response to the previous reviews.

**Reviewer #1 (Public Review):**
Summary:The study of human intelligence has been the focus of cognitive neuroscience research, and finding some objective behavioral or neural indicators of intelligence has been an ongoing problem for scientists for many years. Melnick et al, 2013 found for the first time that the phenomenon of spatial suppression in motion perception predicts an individual's IQ score. This is because IQ is likely associated with the ability to suppress irrelevant information. In this study, a high-resolution MRS approach was used to test this theory. In this paper, the phenomenon of spatial suppression in motion perception was found to be correlated with the visuo-spatial subtest of gF, while both variables were also correlated with the GABA concentration of MT+ in the human brain. In addition, there was no significant relationship with the excitatory transmitter Glu. At the same time, SI was also associated with MT+ and several frontal cortex FCs.Strengths:(1) 7T high-resolution MRS is used.(2) This study combines the behavioral tests, MRS, and fMRI.Weaknesses:Major:In Melnick (2013) IQ scores were measured by the full set of WAIS-III, including all subtests. However, this study only used visual spatial domain of gF. I wonder why only the visuo-spatial subtest was used not the full WAIS-III? I am wondering whether other subtests were conducted and, if so, please include the results as well to have comprehensive comparisons with Melnick (2013).

We thank the reviewer for pointing this out. The decision was informed by Melnick’s findings which indicated high correlations between Surround suppression (SI) and the Verbal Comprehension, Perceptual Reasoning, Working Memory, and Processing Speed Indexes, with correlation coefficients of 0.69, 0.47, 0.49, and 0.50, respectively. It is well-established that the hMT+ region of the brain is a sensory cortex involved in visual perception processing (3D perception). Furthermore, motion surround suppression (SI), a specific function of hMT+, aligns closely with this region's activities. Given this context, the Perception Reasoning sub-ability was deemed to have the clearest mechanism for further exploration. Consequently, we selected the most representative subtest of Perception Reasoning—the Block Design Test—which primarily assesses 3D visual intelligence.” For further clarification, due to these reasons, we conducted only the visuo-spatial subtest.

Minor:Comments:In the first revised version, we addressed the following recommendations in the 'Author response' file titled 'Recommendation for the authors.' It seems our response may not have reached you successfully. We would like to share and expand upon our response here:(1) Table 1 and Table supplementary 1-3 contain many correlation results. But what are the main points of these values? Which values do the authors want to highlight? Why are only p-values shown with significance symbols in Table supplementary 2??(1.1) What are the main points of these values?

Thank reviewer for pointing this out. These correlations represent the relationship between behavior task (SI/BDT) and resting-state functional connectivity. It indicates that left hMT+ is involved in the efficient information integration network when it comes to BDT task. In addition, left hMT+’s surround suppression is involved in several hMT+ - frontal connectivity. Furthermore, the overlap regions between two task indicates the underlying mechanism.

(1.2) Which values do the authors want to highlight?

Table 1 and Table Supplementary 1-3 present the preliminary analysis results for Table 2 and Table Supplementary 4-6. So, we generally report all value. Conversely, in the Table 2 and Table Supplementary 4-6, we highlight the value which support our main conclusion.

(1.3) Why are only p-values shown with significance symbols in Table Supplementary 2?

Thank you for pointing this out, it is a mistake. We have revised it and delete the significance symbols.

(2) Line 27, it is unclear to me what is "the canonical theory".

We thank reviewer for pointing this out. We have revised “the canonical theory" to “the prevailing opinion” (line 27)

(3) Throughout the paper, the authors use "MT+", I would suggest using "hMT+" to indicate the human MT complex, and to be consistent with the human fMRI literature.

We thank reviewer for pointing this out. We have revised them.

(4) At the beginning of the results section, I suggest including the total number of subjects. It is confusing what "31/36 in MT+, and 28/36 in V1" means.

We thank reviewer for pointing this out. We have included the total number of subjects in the beginning of result section. (line 110, line 128)

(5) Line 138, "This finding supports the hypothesis that motion perception is associated with neural activity in MT+ area". This sentence is strange because it is a well-established finding in numerous human fMRI papers. I think the authors should be more specific about what this finding implies.

We thank reviewer for pointing this out. We have revised it to:” This finding is in line with prior results, which indicates that motion perception is associated with neural activity in hMT+ area, but not in EVC (primarily in V1)” (lines 156-158)

(6) There are no unit labels for all x- and y-axies in Figure 1. I only see the unit for Conc is mmol per kg wet weight.

We thank reviewer for pointing this out. Figure 1 is a schematic and workflow chart, so labels for x- and y-axes are not needed. I believe this confusion might pertain to Figure 3. In Figures 3a and 3b, the MRS spectrum does not have a standard y-axis unit as it varies based on the individual physical conditions of the scanner; it is widely accepted that no y-axis unit is used. While the x-axis unit is ppm, which indicate the chemical shift of different metabolites. In Figure 3c, the BDT represents IQ scores, which do not have a standard unit. Similarly, in Figures 3d and 3e, the Suppression Index does not have a standard unit.

(7) Although the correlations are not significant in Figure Supplement 2&3, please also include the correlation line, 95% confidence interval, and report the r values and p values (i.e., similar format as in Figure 1C).

We thank reviewer for pointing this out. We have revised them and include the correlation line, 95% confidence interval, r values and p values.

(8) There is no need to separate different correlation figures into Figure Supplementary 1-4. They can be combined into the same figure.

We thank reviewer for the suggestion. However, each correlation figure in the supplementary figures has its own specific topic and conclusion. Please notes that in the revised version, we have added a figure showing the EVC (primarily in V1) MRS scanning ROI as Supplementary Figure 1. Therefore, the figures the reviewer is concerned about are Supplementary Figure 2-5. The correlation figures in Supplementary Figure 2 indicate that GABA in EVC (primarily in V1) does not show any correlation with BDT and SI, illustrating that inhibition in EVC (primarily in V1) is unrelated to both 3D visuo-spatial intelligence and motion suppression processing. The correlations in Supplementary Figure 3 indicate that the excitation mechanism, represented by Glutamate concentration, does not contribute to 3D visuo-spatial intelligence in either hMT+ or EVC (primarily in V1). Supplementary Figure 4 validates our MRS measurements. Supplementary Figure 5 addresses potential concerns regarding the impact of outliers on correlation significance. Even after excluding two “outliers” from Figures 3d and 3e, the correlation results remain stable.

(9) Line 213, as far as I know, the study (Melnick et al., 2013) is a psychophysical study and did not provide evidence that the spatial suppression effect is associated with MT+.

We thank reviewer for pointing this out. It was a mistake to use this reference, and we have revised it accordingly. (line 242)

(10) At the beginning of the results, I suggest providing more details about the motion discrimination tasks and the measurement of the BDT.

We thank reviewer for pointing this out. We have included some brief description of task in the beginning of result section. (lines 116-120)

(11) Please include the absolute duration thresholds of the small and large sizes of all subjects in Figure 1.

We thank reviewer for the suggestion. We have included these results in Figure 3.

(12) Figure 5 is too small. The items in plot a and b can be barely visible.

We thank reviewer for pointing this out. We increase the size and resolution of the Figure.

**Reviewer #3 (Public Review):**
(1) Throughout the manuscript, hMT+ connectivity with the frontal cortex has been treated as an a priori hypothesis/space. However, there is no such motivation or background literature mentioned in the Introduction. Can the authors clarify the necessity of functional connectivity? In other words, can BOLD activity of hMT+ in the localizer task substitute for functional connectivity between hMT+ and the frontal cortex?(1.1) Throughout the manuscript, hMT+ connectivity with the frontal cortex has been treated as an a priori hypothesis/space. However, there is no such motivation or background literature mentioned in the Introduction. Can the authors clarify the necessity of functional connectivity?

We thank reviewer for pointing this out. We offered additional motivation and background literature in the introduction: “Frontal cortex is usually recognized as the cognitive core region (Duncan et al., 2000; Gray et al., 2003). Strong connectivity between the cognitive regions suggests a mechanism for large-scale information exchange and integration in the brain (Barbey, 2018; Cole et al., 2012). Therefore, the potential conjunctive coding may overlap with the inhibition and/or excitation mechanism of hMT+. Taken together, we hypothesized that 3D visuo-spatial intelligence (as measured by BDT) might be predicted by the inhibitory and/or excitation mechanisms in hMT+ and the integrative functions connecting hMT+ with frontal cortex (Figure 1a).” (lines 67-74). Additionally, we have included a whole-brain analysis for validation. Functional connectivity reveals the information exchange relationships across regions, enhancing our understanding of how hMT+ and the frontal cortex collaborate when solving visual-spatial intelligence tasks.

(1.2) In other words, can BOLD activity of hMT+ in the localizer task substitute for functional connectivity between hMT+ and the frontal cortex?

We thank the reviewer for this question. The localizer task was used solely for defining the hMT+ MRS scanning area. Functional connectivity was measured using resting-state fMRI. Research has shown that resting-state functional connectivity between the frontal cortex and other ROIs can further reveal the neural mechanisms underlying intelligence tasks (Song et al., 2008).

(2) There is an obvious mismatch between the in-text description and the content of the figure:"In contrast, there was no correlation between BDT and GABA levels in V1 voxels (figure supplement 1a). Further, we show that SI significantly correlates with GABA levels in hMT+ voxels (r = 0.44, P = 0.01, n = 31, Figure 3d). In contrast, no significant correlation between SI and GABA concentrations in V1 voxels was observed (figure supplement 1b)."

We thank reviewer for pointing this out. We have revised it. The revised version is :” In contrast, there was no correlation between BDT and GABA levels in V1 voxels (figure supplement 2a). Further, we show that SI significantly correlates with GABA levels in hMT+ voxels (r = 0.44, P = 0.01, n = 31, Figure 3d). In contrast, no significant correlation between SI and GABA concentrations in V1 voxels was observed (figure supplement 2b).” (lines 151-156)

(3) The authors' response to my previous round of review indicated that the "V1 ROIs" covered a substantial amount of V3 (32%). Therefore, it would no longer be appropriate to call these "V1 ROIs". I'd suggest renaming them as "Early Visual Cortex (EVC) ROIs" to be more accurate. Can the authors justify why choosing the left hemisphere for visual intelligence task, which is typically believed to be right lateralized?(3.1) The authors' response to my previous round of review indicated that the "V1 ROIs" covered a substantial amount of V3 (32%). Therefore, it would no longer be appropriate to call these "V1 ROIs". I'd suggest renaming them as "Early Visual Cortex (EVC) ROIs" to be more accurate.

We thank the reviewer for pointing this out. We have revised our description of the MRS scanning ROIs to Early Visual Cortex (EVC). Since the majority of our EVC ROIs are in V1 (around 70%) and almost no V2 was included, we decided to mark the EVC ROIs with the explanation "primarily in V1" for better clarification. This terminology has been widely used to better emphasize the V1-based experimental design.

(3.2) Can the authors justify why choosing the left hemisphere for visual intelligence task, which is typically believed to be right lateralized?

We thank the reviewer for pointing this out. The use of the left MT/V5 as a target was motivated by studies demonstrating that left MT+/V5 TMS is more effective at causing perceptual effects (Tadin et al., 2011). Therefore, we chose to use the left hMT+ as our MRS ROI and maintain consistency across different models' ROIs. Additionally, our results support the notion that the visual intelligence task is right lateralized in the frontal cortex. At the resting-fMRI level, we found that significant ROIs, where functional connectivity is highly correlated with BDT scores, are in the right frontal cortex (Figure 5a, b).

(4) "Small threshold" and "large threshold" are neither standard descriptions, and it is unclear what "small threshold" refers to in the following figure caption. Additionally, the unit (ms) is confusing. Does it refer to timing?"(f) Peason's correlation showing significant negative correlations between BDT and small threshold."

Thank you for pointing this out; we agree with your suggestion. We have revised the terms “small threshold” and “large threshold” to “duration threshold of small grating” and “duration threshold of large grating”, respectively. The unit (ms) refers to timing. The details are described in the methods section: “The duration was adaptively adjusted in each trial, and duration thresholds were estimated using a staircase procedure. Thresholds for large and small gratings were obtained from a 160-trial block that contained four interleaved 3-down/1-up staircases. For each participant, we computed the correct rate for different stimulus durations separately for each stimulus size. These values were then fitted to a cumulative Gaussian function, and the duration threshold corresponding to the 75% correct point on the psychometric function was estimated for each stimulus size”.

(5) In the response letter, the authors mentioned incorporating the neural efficiency hypothesis in the Introduction, but the revised Introduction does not contain such information.

We thank the reviewer for pointing this out. In our revised version, the second paragraph of the introduction addresses the neural efficiency hypothesis: “The “neuro-efficiency” hypothesis is one explanation for individual differences in gF (Haier et al., 1988). This hypothesis puts forward that the human brain’s ability to suppress irrelevant information leads to more efficient cognitive processing. Correspondingly, using a well-known visual motion paradigm (center-surround antagonism) (Liu et al., 2016; Tadin et al., 2003), Melnick et al found a strong link between suppression index (SI) of motion perception and the scores of the block design test BDT, a subtest of the Wechsler Adult Intelligence Scale (WAIS), which measures the visuo-spatial component (3D domain) of gF (Melnick et al., 2013). Motion surround suppression (SI), a specific function of human extrastriate cortical region, middle temporal complex (hMT+), aligns closely with this region's activities (Gautama & Van Hulle, 2001). Furthermore, hMT+ is a sensory cortex involved in visual perception processing (3D domain) (Cumming & DeAngelis, 2001). These findings suggest that hMT+ potentially plays a significant role in 3D visuo-spatial intelligence by facilitating the efficient processing of 3D visual information and suppressing irrelevant information. However, more evidence is needed to uncover how the hMT+ functions as a core region for 3D visuo-spatial intelligence.” (lines 51-66)

**Recommendations for the authors:**

**Reviewer #1 (Recommendations for The Authors):**
In the Code availability, it states that "this paper does not report original code". It seems weird because at least the code to reproduce the figures from the data should be provided.

Thank you for pointing this out. Almost all figures were created using software such as DPABI, BrainNet, and GraphPad Prism 9.5, which are manually operated and do not require code adjustments. However, for the MRS fitting curve, we can provide our MATLAB code for redrawing the MRS fitting. The code has been uploaded to Zenodo.